# Improving Exploration in Evolution Strategies for Deep Reinforcement Learning via a Population of Novelty-Seeking Agents

**Edoardo Conti**[*]     **Vashisht Madhavan**[*]     **Felipe Petroski Such**
**Joel Lehman**     **Kenneth O. Stanley**     **Jeff Clune**
Uber AI Labs

## Abstract

Evolution strategies (ES) are a family of black-box optimization algorithms able to train deep neural networks roughly as well as Q-learning and policy gradient methods on challenging deep reinforcement learning (RL) problems, but are much faster (e.g. hours vs. days) because they parallelize better. However, many RL problems require directed exploration because they have reward functions that are sparse or deceptive (i.e. contain local optima), and it is unknown how to encourage such exploration with ES. Here we show that algorithms that have been invented to promote directed exploration in small-scale evolved neural networks via populations of exploring agents, specifically novelty search (NS) and quality diversity (QD) algorithms, can be hybridized with ES to improve its performance on sparse or deceptive deep RL tasks, while retaining scalability. Our experiments confirm that the resultant new algorithms, NS-ES and two QD algorithms, NSR-ES and NSRA-ES, avoid local optima encountered by ES to achieve higher performance on Atari and simulated robots learning to walk around a deceptive trap. This paper thus introduces a family of fast, scalable algorithms for reinforcement learning that are capable of directed exploration. It also adds this new family of exploration algorithms to the RL toolbox and raises the interesting possibility that analogous algorithms with multiple simultaneous paths of exploration might also combine well with existing RL algorithms outside ES.

## 1   Introduction

In RL, an agent tries to learn to perform a sequence of actions in an environment that maximizes some notion of cumulative reward [1]. However, reward functions are often *deceptive*, and solely optimizing for reward without some mechanism to encourage intelligent exploration can lead to getting stuck in local optima and the agent failing to properly learn [1–3]. Unlike in supervised learning with deep neural networks (DNNs), wherein local optima are not thought to be a problem [4, 5], the training data in RL is determined by the actions an agent takes. If the agent greedily takes actions that maximize reward, the training data for the algorithm will be limited and it may not discover alternate strategies with larger payoffs (i.e. it can get stuck in local optima) [1–3]. Sparse reward signals can also be a problem for algorithms that only maximize reward, because at times there may be no reward gradient to follow. The possibility of deceptiveness and/or sparsity in the reward signal motivates the need for efficient and *directed* exploration, in which an agent is motivated to visit unexplored states in order to learn to accumulate higher rewards. Although deep RL algorithms have performed amazing feats in recent years [6–8], they have mostly done so despite relying on simple, *undirected* (aka dithering) exploration strategies, in which an agent hopes to explore new areas of its environment by taking random actions (e.g. epsilon-greedy exploration) [1].

A number of methods have been proposed to promote directed exploration in RL [9, 10], including recent methods that handle high-dimensional state spaces with DNNs. A common idea is to encourage

---

[*]Equal contribution, corresponding authors: `vashisht@uber.com`, `edoardo.conti@gmail.com`.

an agent to visit states it has rarely or never visited (or take novel actions in those states). Methods proposed to track state (or state-action pair) visitation frequency include (1) approximating state visitation counts based on either auto-encoded latent codes of states [11] or pseudo-counts from state-space density models [12, 13], (2) learning a dynamics model that predicts future states (assuming predictions will be worse for rarely visited states/state-action pairs) [14–16], and (3) methods based on compression (novel states should be harder to compress) [9].

Those methods all count each state separately. A different approach to is to hand-design (or learn) an abstract, holistic description of an agent's lifetime behavior, and then encourage the agent to exhibit different behaviors from those previously performed. That is the approach of novelty search (NS) [3] and quality diversity (QD) algorithms [17–19], which are described in detail below. Such algorithms are also interestingly different, and have different capabilities, because they perform exploration with a population of agents rather than a single agent (discussed in SI Sec. 6.2). NS and QD have shown promise with smaller neural networks on problems with low-dimensional input and output spaces [17–22]. Evolution strategies (ES) [23] was recently shown to perform well on high-dimensional deep RL tasks in a short amount of wall clock time by scaling well to many distributed computers. In this paper, for the first time, we study how these two types of algorithms can be hybridized with ES to scale them to deep neural networks and thus tackle hard, high-dimensional deep reinforcement learning problems, without sacrificing the speed/scalability benefits of ES. We first study NS, which performs exploration *only* (ignoring the reward function) to find a set of novel solutions [3]. We then investigate algorithms that balance exploration and exploitation, specifically novel instances of QD algorithms, which seek to produce a set of solutions that are both novel and high-performing [17–20]. Both NS and QD are explained in detail in Sec. 3.

ES directly searches in the parameter space of a neural network to find an effective policy. A team from OpenAI recently showed that ES can achieve competitive performance on many reinforcement learning (RL) tasks while offering some unique benefits over traditional gradient-based RL methods [24]. Most notably, ES is highly parallelizable, which enables near linear speedups in runtime as a function of CPU/GPU workers. For example, with hundreds of parallel CPUs, ES was able to achieve roughly the same performance on Atari games with the same DNN architecture in 1 hour as A3C did in 24 hours [24]. In this paper, we investigate adding NS and QD to ES only; in future work, we will investigate how they might be hybridized with Q-learning and policy gradient methods. We start with ES because (1) its fast wall-clock time allows rapid experimental iteration, and (2) NS and QD were originally developed as neuroevolution methods, making it natural to try them first with ES, which is also an evolutionary algorithm.

Here we test whether encouraging novelty via NS and QD improves the performance of ES on sparse and/or deceptive control tasks. Our experiments confirm that NS-ES and two simple versions of QD-ES (NSR-ES and NSRA-ES) avoid local optima encountered by ES and achieve higher performance on tasks ranging from simulated robots learning to walk around a deceptive trap to the high-dimensional pixel-to-action task of playing Atari games. Our results add these new families of exploration algorithms to the RL toolbox, opening up avenues for studying how they can best be combined with RL algorithms, whether ES or others.

## 2 Background

### 2.1 Evolution Strategies

Evolution strategies (ES) are a class of black box optimization algorithms inspired by natural evolution [23]: At every iteration (generation), a population of parameter vectors (genomes) is perturbed (mutated) and, optionally, recombined (merged) via crossover. The fitness of each resultant offspring is then evaluated according to some objective function (reward) and some form of selection then ensures that individuals with higher reward tend to produce offspring for the next generation. Many algorithms in the ES class differ in their representation of the population and methods of recombination; the algorithms subsequently referred to in this work belong to the class of Natural Evolution Strategies (NES) [25, 26]. NES represents the population as a distribution of parameter vectors $\theta$ characterized by parameters $\phi$: $p_\phi(\theta)$. Under a fitness function, $f(\theta)$, NES seeks to maximize the average fitness of the population, $\mathbb{E}_{\theta \sim p_\phi}[f(\theta)]$, by optimizing $\phi$ with stochastic gradient ascent.

Recent work from OpenAI outlines a version of NES applied to standard RL benchmark problems [24]. We will refer to this variant simply as ES going forward. In their work, a fitness function $f(\theta)$ represents the stochastic reward experienced over a full episode of agent interaction, where $\theta$

parameterizes the policy $\pi_\theta$. From the population distribution $p_{\phi_t}$, parameters $\theta_t^i \sim \mathcal{N}(\theta_t, \sigma^2 I)$ are sampled and evaluated to obtain $f(\theta_t^i)$. In a manner similar to REINFORCE [27], $\theta_t$ is updated using an estimate of approximate gradient of expected reward:

$$\nabla_\phi \mathbb{E}_{\theta \sim \phi}[f(\theta)] \approx \frac{1}{n} \sum_{i=1}^n f(\theta_t^i) \nabla_\phi \log p_\phi(\theta_t^i)$$

where $n$ is the number of samples evaluated per generation. Intuitively, NES samples parameters in the neighborhood of $\theta_t$ and determines the direction in which $\theta_t$ should move to improve expected reward. Since this gradient estimate has high variance, NES relies on a large $n$ for variance reduction. Generally, NES also evolves the covariance of the population distribution, but for the sake of fair comparison with Salimans et al. [24] we consider only static covariance distributions, meaning $\sigma$ is fixed throughout training.

To sample from the population distribution, Salimans et al. [24] apply additive Gaussian noise to the current parameter vector : $\theta_t^i = \theta_t + \sigma \epsilon_i$ where $\epsilon_i \sim \mathcal{N}(0, I)$. Although $\theta$ is high-dimensional, previous work has shown Gaussian parameter noise to have beneficial exploration properties when applied to deep networks [26, 28, 29]. The gradient is then estimated by taking a sum of sampled parameter perturbations weighted by their reward:

$$\nabla_{\theta_t} \mathbb{E}_{\epsilon \sim \mathcal{N}(0, I)}[f(\theta_t + \sigma \epsilon)] \approx \frac{1}{n\sigma} \sum_{i=1}^n f(\theta_t^i) \epsilon_i$$

To ensure that the scale of reward between domains does not bias the optimization process, we follow the approach of Salimans et al. [24] and rank-normalize $f(\theta_t^i)$ before taking the weighted sum. Overall, this NES variant exhibits performance on par with contemporary, gradient-based algorithms on difficult RL domains, including simulated robot locomotion and Atari environments [30].

## 2.2 Novelty Search (NS)

Inspired by nature's drive towards diversity, NS encourages policies to engage in notably different behaviors than those previously seen. The algorithm encourages different behaviors by computing the *novelty* of the current policy with respect to previously generated policies and then encourages the population distribution to move towards areas of parameter space with high novelty. NS outperforms reward-based methods in maze and biped walking domains, which possess deceptive reward signals that attract agents to local optima [3]. In this work, we investigate the efficacy of NS at the scale of DNNs by combining it with ES. In NS, a policy $\pi$ is assigned a domain-dependent *behavior characterization* $b(\pi)$ that describes its behavior. For example, in the case of a humanoid locomotion problem, $b(\pi)$ may be as simple as a two-dimensional vector containing the humanoid's final $\{x, y\}$ location. Throughout training, every $\pi_\theta$ evaluated adds $b(\pi_\theta)$ to an archive set $A$ with some probability. A particular policy's novelty $N(b(\pi_\theta), A)$ is then computed by selecting the k-nearest neighbors of $b(\pi_\theta)$ from $A$ and computing the average distance between them:

$$N(\theta, A) = N(b(\pi_\theta), A) = \frac{1}{|S|} \sum_{j \in S} ||b(\pi_\theta) - b(\pi_j)||_2$$

$$S = kNN(b(\pi_\theta), A)$$

$$= \{b(\pi_1), b(\pi_2), ..., b(\pi_k)\}$$

Above, the distance between behavior characterizations is calculated with an $L2$-norm, but any distance function can be substituted. Previously, NS has been implemented with a genetic algorithm [3]. We next explain how NS can now be combined with ES, to leverage the advantages of both.

## 3 Methods

### 3.1 NS-ES

We use the ES optimization framework, described in Sec. 2.1, to compute and follow the gradient of expected novelty with respect to $\theta_t$. Given an archive $A$ and sampled parameters $\theta_t^i = \theta_t + \sigma \epsilon_i$, the gradient estimate can be computed:

$$\nabla_{\theta_t} \mathbb{E}_{\epsilon \sim \mathcal{N}(0, I)}[N(\theta_t + \sigma \epsilon, A)|A] \approx \frac{1}{n\sigma} \sum_{i=1}^n N(\theta_t^i, A) \epsilon_i$$

The gradient estimate obtained tells us how to change the current policy's parameters $\theta_t$ to increase the average novelty of our parameter distribution. We condition the gradient estimate on A, as the archive is fixed at the beginning of a given iteration and updated only at the end. We add only the behavior characterization corresponding to each $\theta_t$, as adding those for each sample $\theta_t^i$ would inflate the archive and slow the nearest-neighbors computation. As more behavior characterizations are added to $A$, the novelty landscape changes, resulting in commonly occurring behaviors becoming "boring." Optimizing for expected novelty leads to policies that move towards unexplored areas of behavior space.

NS-ES could operate with a single agent that is rewarded for acting differently than its ancestors. However, to encourage additional diversity and get the benefits of population-based exploration described in SI Sec. 6.2, we can instead create a population of $M$ agents, which we will refer to as the *meta-population*. Each agent, characterized by a unique $\theta^m$, is rewarded for being different from all prior agents in the archive (ancestors, other agents, and the ancestors of other agents), an idea related to that of Liu et al. [31], which optimizes for a distribution of M diverse, high-performing policies. We hypothesize that the selection of $M$ is domain dependent and that identifying which domains favor which regime is a fruitful area for future research.

We initialize $M$ random parameter vectors and at every iteration select one to update. For our experiments, we probabilistically select which $\theta^m$ to advance from a discrete probability distribution as a function of $\theta^m$'s novelty. Specifically, at every iteration, for a set of agent parameter vectors $\Pi = \{\theta^1, \theta^2, ..., \theta^M\}$, we calculate each $\theta^m$'s probability of being selected $P(\theta^m)$ as its novelty normalized by the sum of novelty across all policies:

$$P(\theta^m) = \frac{N(\theta^m, A)}{\sum_{j=1}^{M} N(\theta^j, A)} \tag{1}$$

Having multiple, separate agents represented as independent Gaussians is a simple choice for the *meta-population* distribution. In future work, more complex sampling distributions that represent the multi-modal nature of meta-population parameter vectors could be tried.

After selecting an individual $m$ from the meta-population, we compute the gradient of expected novelty with respect to $m$'s current parameter vector, $\theta_t^m$, and perform an update step accordingly:

$$\theta_{t+1}^m \leftarrow \theta_t^m + \alpha \frac{1}{n\sigma} \sum_{i=1}^{n} N(\theta_i^{i,m}, A) \epsilon_i$$

Where $n$ is the number of sampled perturbations to $\theta_t^m$, $\alpha$ is the stepsize, and $\theta_i^{i,m} = \theta_t^m + \sigma \epsilon_i$, where $\epsilon_i \sim \mathcal{N}(0, I)$. Once the current parameter vector is updated, $b(\pi_{\theta_{t+1}^m})$ is computed and added to the shared archive $A$. The whole process is repeated for a pre-specified number of iterations, as there is no true convergence point of NS. During training, the algorithm preserves the policy with the highest average episodic reward and returns this policy once training is complete. Although Salimans et al. [24] return only the final policy after training with ES, the ES experiments in this work return the best-performing policy to facilitate fair comparison with NS-ES. Algorithm 1 in SI Sec. 6.5 outlines a simple, parallel implementation of NS-ES. It is important to note that the addition of the archive and the replacement of the fitness function with novelty does not damage the scalability of the ES optimization procedure (SI Sec. 6.4).

### 3.2 QD-ES Algorithms: NSR-ES and NSRA-ES

NS-ES alone can enable agents to avoid deceptive local optima in the reward function. Reward signals, however, are still very informative and discarding them completely may cause performance to suffer. Consequently, we train a variant of NS-ES, which we call NSR-ES, that combines the reward ("fitness") and novelty calculated for a given set of policy parameters $\theta$. Similar to NS-ES and ES, NSR-ES operates on entire episodes and can thus evaluate reward and novelty simultaneously for any sampled parameter vector: $\theta_t^{i,m} = \theta_t^m + \epsilon_i$. Specifically, we compute $f(\theta_t^{i,m})$ and $N(\theta_t^{i,m}, A)$, average the two values, and set the average as the weight for the corresponding $\epsilon_i$. The averaging process is integrated into the parameter update rule as:

$$\theta_{t+1}^m \leftarrow \theta_t^m + \alpha \frac{1}{n\sigma} \sum_{i=1}^{n} \frac{f(\theta_t^{i,m}) + N(\theta_t^{i,m}, A)}{2} \epsilon_i$$

Intuitively, the algorithm follows the approximated gradient in parameter-space towards policies that both exhibit novel behaviors and achieve high rewards. Often, however, the scales of $f(\theta)$ and $N(\theta, A)$ differ. To combine the two signals effectively, we rank-normalize $f(\theta_t^{i,m})$ and $N(\theta_t^{i,m}, A)$ independently before computing the average. Optimizing a linear combination of novelty and reward was previously explored in Cuccu and Gomez [32] and Cuccu et al. [33], but not with large neural networks on high-dimensional problems. The result of NSR-ES is a set of $M$ agents being optimized to be both high-performing, yet different from each other.

NSR-ES has an equal weighting of the performance and novelty gradients that is static across training. We explore a further extension of NSR-ES called NSRAdapt-ES (NSRA-ES), which takes advantage of the opportunity to dynamically weight the priority given to the performance gradient $f(\theta_t^{i,m})$ vs. the novelty gradient $N(\theta_t^{i,m}, A)$ by intelligently adapting a weighting parameter $w$ during training. By doing so, the algorithm can follow the performance gradient when it is making progress, increasingly

try different things if stuck in a local optimum, and switch back to following the performance gradient once unstuck. For a specific $w$ at a given generation, the parameter update rule for NSRA-ES is expressed as follows:

$$\theta_{t+1}^m \leftarrow \theta_t^m + \alpha \frac{1}{n\sigma} \sum_{i=1}^n w f(\theta_t^{i,m}) \epsilon_i + (1-w) N(\theta_t^{i,m}, A) \epsilon_i$$

We set $w = 1.0$ initially and decrease it if performance stagnates across a fixed number of generations. We continue decreasing $w$ until performance increases, at which point we increase $w$. While many previous works have adapted exploration pressure online by learning the amount of noise to add to the parameters [25, 26, 28, 34], such approaches rest on the assumption that an increased amount of parameter noise leads to increased *behavioral diversity*, which is often not the case (e.g. too much noise may lead to degenerate policies) [20]. Here we directly adapt the weighting between behavioral diversity and performance, which more directly controls the trade-off of interest. SI Sec. 6.5 provides a more detailed description of how we adapt $w$ as well as pseudocode for NSR-ES and NSRA-ES. Source code and hyperparameter settings for our experiments can be found here: https://github.com/uber-research/deep-neuroevolution

# 4 Experiments

## 4.1 Simulated Humanoid Locomotion problem

We first tested our implementation of NS-ES, NSR-ES, and NSRA-ES on the problem of having a simulated humanoid learn to walk. We chose this problem because it is a challenging continuous control benchmark where most would presume a reward function is necessary to solve the problem. With NS-ES, we test whether searching through novelty alone can find solutions to the problem. A similar result has been shown for much smaller neural networks ($\sim$50-100 parameters) on a more simple simulated biped [20], but here we test whether NS-ES can enable locomotion at the scale of deep neural networks on a much more sophisticated environment. NSR-ES and NSRA-ES experiments then test the effectiveness of combining exploration and reward pressures on this difficult continuous control problem. SI Sec. 6.7 outlines complete experimental details.

The first environment is in a slightly modified version of OpenAI Gym's Humanoid-v1 environment. Because the heart of this challenge is to learn to walk efficiently, not to walk in a particular direction, we modified the environment reward to be isotropic (i.e. indifferent to the direction the humanoid traveled) by setting the velocity component of reward to distance traveled from the origin as opposed to distance traveled in the positive $x$ direction.

As described in section 2.2, novelty search requires a domain-specific behavior characterization (BC) for each policy, which we denote as $b(\pi_{\theta_i})$. For the Humanoid Locomotion problem the BC is the agent's final $\{x, y\}$ location, as it was in Lehman and Stanley [20]. NS also requires a distance function between two BCs. Following Lehman and Stanley [20], the distance function is the square of the Euclidean distance:

$$dist(b(\pi_{\theta_i}), b(\pi_{\theta_j})) = ||b(\pi_{\theta_i}) - b(\pi_{\theta_j})||_2^2$$

The first result is that ES obtains a higher final reward than NS-ES ($p < 0.05$) and NSR-ES ($p < 0.05$); these and all future $p$ values are calculated via a Mann-Whitney U test. The performance gap is more pronounced for smaller amounts of computation (Fig. 1(c)). However, many will be surprised that NS-ES is still able to consistently solve the problem despite ignoring the environment's reward function. While the BC is *aligned* [35] with the problem in that reaching new $\{x, y\}$ positions tends to also encourage walking, there are many parts of the reward function that the BC ignores (e.g. energy-efficiency, impact costs).

We hypothesize that with a sophisticated BC that encourages diversity in all of the behaviors the multi-part reward function cares about, there would be no performance gap. However, such a BC may be difficult to construct and would likely further exaggerate the amount of computation required for NS to match ES. NSR-ES demonstrates faster learning than NS-ES due to the addition of reward pressure, but ultimately results in similar final performance after 600 generations ($p > 0.05$, Fig. 1 (c)). Promisingly, on this non-deceptive problem, NSRA-ES does not pay a cost for its latent exploration capabilities and performs similarly to ES ($p > 0.05$).

The Humanoid Locomotion problem does not appear to be a deceptive problem, at least for ES. To test whether NS-ES, NSR-ES, and NSR-ES specifically help with deception, we also compare ES to these algorithms on a variant of this environment we created that adds a deceptive trap (a local optimum) that must be avoided for maximum performance (Fig. 1 (b)). In this new environment,

a small three-sided enclosure is placed at a short distance in front of the starting position of the humanoid and the reward function is simply the distance traveled in the positive $x$ direction.

Fig. 1 (d) and SI Sec. 6.8 show the reward received by each algorithm and Fig. 2 shows how the algorithms differ qualitatively during search on this problem. In every run, ES gets stuck in the local optimum due to following reward into the deceptive trap. NS-ES is able to avoid the local optimum as it ignores reward completely and instead seeks to thoroughly explore the environment, but doing so also means it makes slow progress according to the reward function. NSR-ES demonstrates superior performance to NS-ES ($p < 0.01$) and ES ($p < 0.01$) as it benefits from both optimizing for reward and escaping the trap via the pressure for novelty. Like ES, NSRA-ES learns to walk into the deceptive trap initially, as it initially is optimizing for reward only. Once stuck in the local optimum, the algorithm continually increases its pressure for novelty, allowing it to escape the deceptive trap and ultimately achieve much higher rewards than NS-ES ($p < 0.01$) and NSR-ES ($p < 0.01$). Based just on these two domains, NSRA-ES seems to be the best algorithm across the board because it can exploit well when there is no deception, add exploration dynamically when there is, and return to exploiting once unstuck. The latter is likely why NSRA-ES outperforms even NSR-ES on the deceptive humanoid locomotion problem.

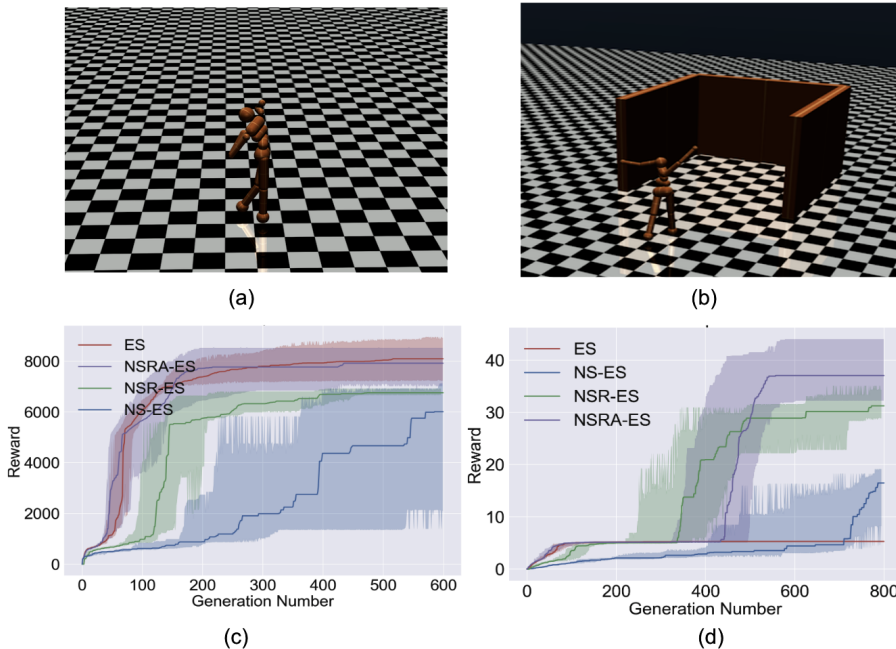

Figure 1: **Humanoid Locomotion Experiment.** The humanoid locomotion task is shown without a deceptive trap (a) and with one (b), and results on them in (c) and (d), respectively. Here and in similar figures below, the median reward (of the best seen policy so far) per generation across 10 runs is plotted as the bold line with 95% bootstrapped confidence intervals of the median (shaded). Following Salimans et al. [24], policy performance is measured as the average performance over ~30 stochastic evaluations.

Fig. 2 also shows the benefit of maintaining a meta-population ($M = 5$) in the NS-ES, NSR-ES, and NSRA-ES algorithms. Some lineages get stuck in the deceptive trap, incentivizing other policies to explore around the trap. At that point, all three algorithms begin to allocate more computational resources to this newly discovered, more promising strategy via the probabilistic selection method outlined in Sec. 3.1. Both the novelty pressure and having a meta-population thus appear to be useful, but in future work we look to disambiguate the relative contribution made by each.

## 4.2 Atari

We also tested NS-ES, NSR-ES, and NSRA-ES on numerous games from the Atari 2600 environment in OpenAI Gym [36]. Atari games serve as an informative benchmark due to their high-dimensional pixel input and complex control dynamics; each game also requires different levels of exploration

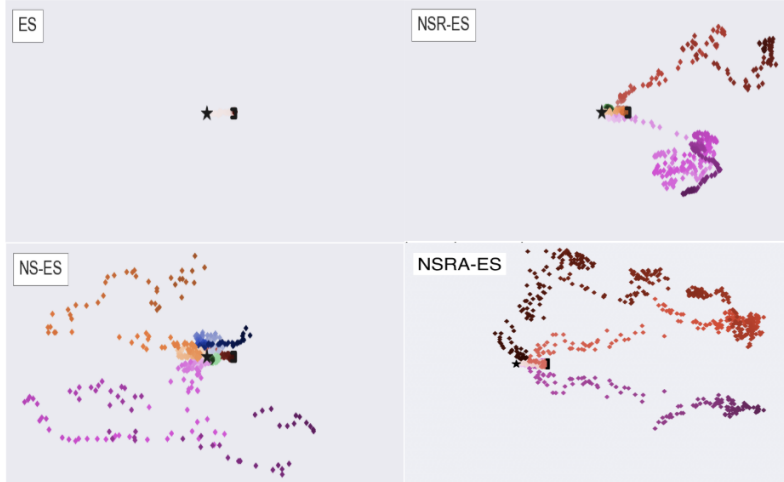

Figure 2: **ES gets stuck in the deceptive local optimum while NS-ES, NSR-ES & NSRA-ES explore to find better solutions.** An overhead view of a representative run is shown for each algorithm on the Humanoid Locomotion with Deceptive Trap problem. The black star represents the humanoid's starting point. Each diamond represents the final location of a generation's policy, i.e. $\pi(\theta_t)$, with darker shading for later generations. For NS-ES, NSR-ES, & NSRA-ES plots, each of the $M = 5$ agents in the meta-population and its descendants are represented by different colors. Similar plots for all 10 runs of each algorithm are provided in SI Sec. 6.10.

to solve. To demonstrate the effectiveness of NS-ES, NSR-ES, and NSRA-ES for local optima avoidance and directed exploration, we tested on 12 different games with varying levels of complexity, as defined by the taxonomy in Bellemare et al. [12]. Primarily, we focused on games in which, during preliminary experiments, we observed ES prematurely converging to local optima (Seaquest, Q*Bert, Freeway, Frostbite, and Beam Rider). However, we also included a few other games where ES did not converge to local optima to understand the performance of our algorithm in less-deceptive domains (Alien, Amidar, Bank Heist, Gravitar, Zaxxon, and Montezuma's Revenge). SI Sec. 6.6 describes additional experimental details. We report the median reward across 5 independent runs of the best policy found in each run (see Table 1).

For the behavior characterization, we follow an idea from Naddaf [37] and concatenate Atari game RAM states for each timestep in an episode. RAM states in Atari 2600 games are integer-valued vectors of length 128 in the range [0, 255] that describe all the state variables in a game (e.g. the location of the agent and enemies). Ultimately, we want to automatically learn behavior characterizations directly from pixels. A plethora of recent research suggests that this is a viable approach [12, 38, 39]. For example, low-dimensional, latent representations of the state space could be extracted from auto-encoders [11, 40] or networks trained to predict future states [14, 16]. In this work, however, we focus on learning with a pre-defined, informative behavior characterization and leave the task of jointly learning a policy and latent representation of states for future work. In effect, basing novelty on RAM states provides a confirmation of what is possible in principle with a sufficiently informed behavior characterization. We also emphasize that, while during training NS-ES, NSR-ES, and NSRA-ES use RAM states to guide novelty search, the policy itself, $\pi_{\theta_t}$, operates only on image input and can be evaluated without any RAM state information. The distance between behavior characterizations is the sum of L2-distances at each timestep $t$:

$$dist(b(\pi_{\theta_i}), b(\pi_{\theta_j})) = \sum_{t=1}^{T} ||(b_t(\pi_{\theta_i})) - b_t(\pi_{\theta_j}))||_2$$

For trajectories of different lengths, the last state of the shorter trajectory is repeated until the lengths of both match. Because the BC distance is not normalized by trajectory length, novelty is biased to be higher for longer trajectories. In some Atari games, this bias can lead to higher performing policies, but in other games longer trajectories tend to have a neutral or even negative relationship with performance. In this work we found it beneficial to keep novelty unnormalized, but further investigation into different BC designs could yield additional improvements.

Table 1 compares the performance of each algorithm discussed above to each other and with those from two popular methods for exploration in RL, namely Noisy DQN [29] and A3C+ [12]. Noisy DQN and A3C+ only outperform all the ES variants considered in this paper on 3/12 games and 2/12 games respectively. NSRA-ES, however, outperforms the other algorithms on 5/12 games, suggesting that NS and QD are viable alternatives to contemporary exploration methods.

While the novelty pressure in NS-ES does help it avoid local optima in some cases (discussed below), optimizing for novelty alone does not result in higher reward in most games (although it does in some). However, it is surprising how well NS-ES does in many tasks given that it is not explicitly attempting to increase reward. Because NSR-ES combines exploration with reward maximization, it is able to avoid local optima encountered by ES while also learning to play the game well. In each of the 5 games in which we observed ES converging to premature local optima (i.e. Seaquest, Q*Bert, Freeway, Beam Rider, Frostbite), NSR-ES achieves a higher median reward. In the other games, ES does not benefit from adding an exploration pressure and NSR-ES performs worse. It is expected that if there are no local optima and reward maximization is sufficient to perform well, the extra cost of encouraging exploration will hurt performance. Mitigating such costs, NSRA-ES optimizes solely for reward until a performance plateau is reached. After that, the algorithm will assign more weight to novelty and thus encourage exploration. We found this to be beneficial, as NSRA-ES achieves higher median rewards than ES on 8/12 games and NSR-ES on 9/12 games. It's superior performance validates NSRA-ES as the best among the evolutionary algorithms considered and suggests that using an *adaptive* weighting between novelty and reward is a promising direction for future research.

In the game Seaquest, the avoidance of local optima is particularly important (Fig. 3). ES performance flatlines early at a median reward of 960, which corresponds to a behavior of the agent descending to the bottom, shooting fish, and never coming up for air. This strategy represents a classic local optima, as coming up for air requires temporarily foregoing reward, but enables far higher rewards to be earned in the long run (Salimans et al. [24] did not encounter this particular local optimum with their hyperparameters, but the point is that ES without exploration can get stuck indefinitely on whichever major local optima it encounters). NS-ES learns to come up for air in all 5 runs and achieves a slightly higher median reward of 1044.5 ($p < 0.05$). NSR-ES also avoids this local optimum, but its additional reward signal helps it play the game better (e.g. it is better at shooting enemies), resulting in a significantly higher median reward of 2329.7 ($p < 0.01$). Because NSRA-ES takes reward steps initially, it falls into the same local optimum as ES. Because we chose (without performing a hyperparameter search) to change the weighting $w$ between performance and novelty infrequently (only every 50 generations), and to change it by a small amount (only 0.05), 200 generations was not long enough to emphasize novelty enough to escape this local optimum. We found that by changing $w$ every 10 generations, this problem is remedied and the performance of NSRA-ES equals that of NSR-ES ($p > 0.05$, Fig. 3). These results motivate future research into better hyperparameters for changing $w$, and into more complex, intelligent methods of dynamically adjusting $w$, including with a population of agents with different dynamic $w$ strategies.

The Atari results illustrate that NS is an effective mechanism for encouraging directed exploration, given an appropriate behavior characterization, for complex, high-dimensional control tasks. A novelty pressure alone produces impressive performance on many games, sometimes even beating ES. Combining novelty and reward performs far better, and improves ES performance on tasks where it appears to get stuck on local optima.

## 5    Discussion and Conclusion

NS and QD are classes of evolutionary algorithms designed to avoid local optima and promote exploration in RL environments, but have only been previously shown to work with small neural networks (on the order of hundreds of connections). ES was recently shown to be capable of training deep neural networks that can solve challenging, high-dimensional RL tasks [24]. It also is much faster when many parallel computers are available. Here we demonstrate that, when hybridized with ES, NS and QD not only preserve the attractive scalability properties of ES, but also help ES explore and avoid local optima in domains with deceptive reward functions. To the best of our knowledge, this paper reports the first attempt at augmenting ES to perform directed exploration in high-dimensional environments. We thus provide an option for those interested in taking advantage of the scalability of ES, but who also want higher performance on domains that have reward functions that are sparse or have local optima. The latter scenario will likely hold for most challenging, real-world domains that machine learning practitioners will wish to tackle in the future.

| GAME | ES | NS-ES | NSR-ES | NSRA-ES | DQN | NOISYDQN | A3C+ |
|---|---|---|---|---|---|---|---|
| ALIEN | 3283.8 | 1124.5 | 2186.2 | **4846.4** | 2404 | 2403 | 1848.3 |
| AMIDAR | 322.2 | 134.7 | 255.8 | 305.0 | 924 | **1610** | 964.7 |
| BANK HEIST | 140.0 | 50.0 | 130.0 | 152.9 | 455 | **1068** | 991.9 |
| BEAM RIDER[†] | 871.7 | 805.5 | 876.9 | 906.4 | 10564 | **20793** | 5992.1 |
| FREEWAY[†] | 31.1 | 22.8 | 32.3 | **32.9** | 31 | 32 | 27.3 |
| FROSTBITE[†] | 367.4 | 250.0 | 2978.6 | **3785.4** | 1000 | 753 | 506.6 |
| GRAVITAR | 1129.4 | 527.5 | 732.9 | **1140.9** | 366 | 447 | 246.02 |
| MONTEZUMA | 0.0 | 0.0 | 0.0 | 0.0 | 2 | 3 | **142.5** |
| MS. PACMAN | 4498.0 | 2252.2 | 3495.2 | **5171.0** | 2674 | 2722 | 2380.6 |
| Q*BERT[†] | 1075.0 | 1234.1 | 1400.0 | 1350.0 | 11241 | 15545 | **15804.7** |
| SEAQUEST[†] | 960.0 | 1044.5 | 2329.7 | 960.0 | **4163** | 2282 | 2274.1 |
| ZAXXON | **9885.0** | 1761.9 | 6723.3 | 7303.3 | 4806 | 6920 | 7956.1 |

Table 1: **Atari Results.** The scores are the median, across 5 runs, of the mean reward (over ∼30 stochastic evaluations) of each run's best policy. SI Sec. 6.9 plots performance over time, along with bootstrapped confidence intervals of the median, for each ES algorithm for each game. In some cases rewards reported here for ES are lower than those in Salimans et al. [24], which could be due to differing hyperparameters (SI Sec. 6.6). Games with a † are those in which we observed ES to converge prematurely, presumably due to it encountering local optima. The DQN and A3C results are reported after 200M frames of, and one to many days of, training. All evolutionary algorithm results are reported after ∼ 2B frames of, and ∼2 hours of, training

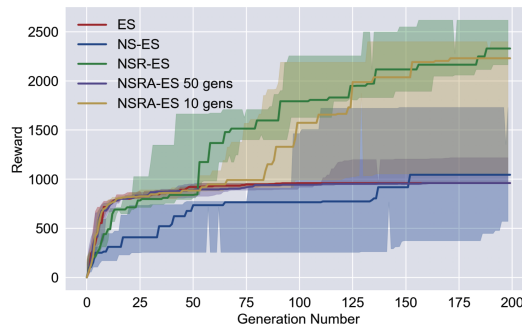

Figure 3: **Seaquest Case Study.** By switching the weighting between novelty and reward, $w$, every 10 generations instead of every 50, NSRA-ES is able to overcome the local optimum ES finds and achieve high scores on Seaquest.

Additionally, this work highlights alternate options for exploration in RL domains. The first is to holistically describe the behavior of an agent instead of defining a per-state exploration bonus. The second is to encourage a population of agents to simultaneously explore different aspects of an environment. These new options thereby open new research areas into (1) comparing holistic vs. state-based exploration, and population-based vs. single-agent exploration, more systematically and on more domains, (2) investigating the best way to combine the merits of all of these options, and (3) hybridizing holistic and/or population-based exploration with other algorithms that work well on deep RL problems, such as policy gradients and DQN. It should be relatively straightforward to combine NS with policy gradients (NS-PG). It is less obvious how to combine it with Q-learning (NS-Q), but may be possible.

As with any exploration method, encouraging novelty can come at a cost if such an exploration pressure is not necessary. In Atari games such as Alien and Gravitar, and in the Humanoid Locomotion problem without a deceptive trap, both NS-ES and NSR-ES perform worse than ES. To avoid this cost, we introduce the NSRA-ES algorithm, which attempts to invest in exploration only when necessary. NSRA-ES tends to produce better results than ES, NS-ES, and NSR-ES across many different domains, making it an attractive new algorithm for deep RL tasks. Similar strategies for adapting the amount of exploration online may also be advantageous for other deep RL algorithms. How best to dynamically balance between exploitation and exploration in deep RL remains an open, critical research challenge, and our work underscores the importance of, and motivates further, such work. Overall, our work shows that ES is a rich and unexploited parallel path for deep RL research. It is worthy of exploring not only because it is an alternative algorithm for RL problems, but also because innovations created in the ES family of algorithms could be ported to improve other deep RL algorithm families like policy gradients and Q learning, or through hybrids thereof.

## Acknowledgments

We thank all of the members of Uber AI Labs, in particular Thomas Miconi, Rui Wang, Peter Dayan, John Sears, Joost Huizinga, and Theofanis Karaletsos, for helpful discussions. We also thank Justin Pinkul, Mike Deats, Cody Yancey, Joel Snow, Leon Rosenshein and the entire OpusStack Team inside Uber for providing our computing platform and for technical support.

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
