[Supplementary Material]

# 6 Supplementary Information

## 6.1 Videos of agent behavior

Videos of example agent behavior in all the environments can be viewed here: `https://goo.gl/cVUG2U`.

## 6.2 Population-based exploration vs. single-agent exploration

As mentioned in the introduction, aside from being holistic vs. state-based, there is another interesting difference between most exploration methods for RL [12–16] and the NS/QD family of algorithms. We do not investigate the benefits of this difference experimentally in this paper, but they are one reason we are interested in NS/QD as an exploration method for RL. One commonality among the previous methods is that the exploration is performed by a *single* agent, a choice that has interesting consequences for learning. To illustrate these consequences, we borrow an example from Stanton and Clune [41]. Imagine a cross-shaped maze (Fig. 4) where to go in each cardinal direction an agent must master a different skill (e.g. going north requires learning to swim, west requires climbing mountains, east requires walking on sand, and south requires walking on ice). Assume rewards may or may not exist at the end of each corridor, so all corridors need to be explored. A single agent has two extreme options, either a depth-first search that serially learns to go to the end of each corridor, or a breadth-first search that goes a bit further in one direction, then comes back to the center and goes a bit further in another direction, etc. Either way, to get to the end of each hallway, the agent will have to at least have traversed each hallway once and thus will have to learn all four sets of skills. With the breadth-first option, all four skillsets must be mastered, but a much longer total distance is traveled.

Figure 4: **Hypothetical Hard Exploration Maze.** In this maze, the agent needs to traverse 4 different terrains to obtain rewards associated with the "?" boxes. Traversing each terrain requires learning a certain skill (i.e. climbing, swimming, etc.). The sprites are from a Super Mario Bros. environment introduced by Paquette [42].

In both cases, another problem arises because, despite recent progress [43, 44], neural networks still suffer from catastrophic forgetting, meaning that as they learn new skills they rapidly lose the ability to perform previously learned ones [45]. Due to catastrophic forgetting, at the end of learning there will be an agent specialized in one of the skills (e.g. swimming), but all of the other skills will have been lost. Furthermore, if any amount of the breadth-first search strategy is employed, exploring each branch a bit further will require relearning that skill mostly from scratch each iteration, significantly slowing exploration. Even if catastrophic forgetting could be solved, there may be limits on the cognitive capacity of single agents (as occurs in humans), preventing one agent from mastering all possible skills.

A different approach is to explore with a *population* of agents. In that case, separate agents could become experts in the separate tasks required to explore in each direction. That may speed learning because each agent can, in parallel, learn only the skills required for its corridor. Additionally, at the end of exploration a specialist will exist with each distinct skill (versus only one skill remaining in the single-agent case). The resulting population of specialists, each with a different skill or way of solving a problem, can then be harnessed by other machine learning algorithms that efficiently search through the repertoire of specialists to find the skill or behavior needed in a particular situation [17, 46]. The

skills of each specialist (in any combination or number) could also then be combined into a single generalist via policy distillation [47]. A further benefit of the population-based approach is, when combining exploration with some notion of quality (e.g. maximizing reward), a population can try out many different strategies/directions and, once one or a few promising strategies are found, the algorithm can reallocate resources to pursue them. The point is not that population-based exploration methods are better or worse than single-agent exploration methods (when holding computation constant), but instead that they are a different option with different capabilities, pros, and cons, and are thus worth investigating [41]. Supporting this view, recent work has demonstrated the benefits of populations for deep learning [48, 49].

### 6.3 Choosing an appropriate behavior characterization

Although optimizing for novelty can be a useful exploration signal for RL agents, the efficacy of optimizing for novelty is determined by the choice of behavior characterization (BC). Often, a BC can be difficult to specify for complex environments, for which the principal axes of effective exploration cannot be enumerated. However, choosing an informative BC, like choosing an informative reward function, is a useful way to inject domain knowledge to induce specific agent behavior. In cases where designing the BC is cumbersome, methods do exist to systematically derive or learn them [50, 51]. The Atari results for NSR-ES and NSRA-ES also suggest that even if the BC is not selected carefully for the domain (the RAM state was not designed to be a BC, and thus being diverse in some RAM state dimensions does not lead to fruitful exploration), novelty with respect to the BC is still a useful exploration signal when combined with reward. Although determining appropriate BCs is out of the scope of this work, we believe it is fruitful to investigate the effect of BC choice on exploration in future work and that doing so may further improve performance.

### 6.4 Preserving scalability

As shown in Salimans et al. [24], ES scales well with the amount of computation available. Specifically, as more CPUs are used, training times reduce almost linearly, whereas DQN and A3C are not amenable to massive parallelization. NS-ES, NSR-ES and NSRA-ES, however, enjoy the same parallelization benefits as ES because they use an almost identical optimization process. The addition of an archive between agents in the meta-population does not hurt scalability because $A$ is only updated after $\theta_t^m$ has been updated. Since $A$ is kept fixed during the calculation of $N(\theta_t^{i,m}, A)$ and $f(\theta_t^{i,m})$ for all $i = 1...n$ perturbations, the coordinator only needs to broadcast $A$ once at the beginning of each generation. In all algorithms, the parameter vector $\theta_t^i$ must be broadcast at the beginning of each generation and since $A$ generally takes up much less memory than the parameter vector, broadcasting both would incur effectively zero extra network overhead. NS-ES, NSR-ES, and NSRA-ES do however introduce an additional computation conducted on the coordinator node. At the start of every generation we must compute the novelty of each candidate $\theta_t^m; m \in \{1, ..., M\}$. For an archive of length $n$ this operation is $O(Mn)$, but since $M$ is small and fixed throughout training this cost is not significant in practice. Additionally, there are methods for keeping the archive small if this computation becomes an issue [52].

### 6.5 NS-ES, NSR-ES, and NSRA-ES Algorithms

For the NSRA-ES algorithm, we introduce 3 new hyperparameters: $w$, $t_w$, $\delta_w$. These quantities determine the agent's preference for novelty or reward at various phases in training. For all of our experiments, $w = 1.0$ initially, meaning that NSRA-ES initially follows the gradient of reward only. If the best episodic reward seen, $f_{best}$, does not increase in $t_w = 50$ generations, $w$ is decreased by $\delta_w = 0.05$ and gradients with respect to the new weighted average of novelty and reward are followed. The process is repeated until a new, higher $f_{best}$ is found, at which point $w$ is increased by $\delta_w$. Intuitively the algorithm follows reward gradients until the increases in episodic reward plateau, at which point the agent is increasingly encouraged to explore. The agent will continue to explore until a promising behavior (i.e. one with higher reward than seen so far) is found. NSRA-ES then increases $w$ to pivot back towards exploitation instead of exploration.

### 6.6 Atari training details

Following Salimans et al. [24], the network architecture for the Atari experiments consists of 2 convolutional layers (16 filters of size 8x8 with stride 4 and 32 filters of size 4x4 with stride 2)

---
**Algorithm 1** NS-ES
---
1: **Input:** learning rate $\alpha$, noise standard deviation $\sigma$, number of policies to maintain $M$, iterations $T$, behavior characterization $b(\pi_\theta)$
2: **Initialize:** $M$ randomly initialized policy parameter vectors $\{\theta_0^1, \theta_0^2, ..., \theta_0^M\}$, archive $A$, number of workers $n$
3: **for** $j = 1$ **to** $M$ **do**
4:     Compute $b(\pi_{\theta_0^j})$
5:     Add $b(\pi_{\theta_0^j})$ to $A$
6: **end for**
7: **for** $t = 0$ **to** $T - 1$ **do**
8:     Sample $\theta_t^m$ from $\{\theta_t^1, \theta_t^2, \ldots, \theta_t^M\}$ via eq. [1]
9:     **for** $i = 1$ **to** $n$ **do**
10:         Sample $\epsilon_i \sim \mathcal{N}(0, \sigma^2 I)$
11:         Compute $\theta_t^{i,m} = \theta_t^m + \epsilon_i$
12:         Compute $b(\pi_{\theta_t^{i,m}})$
13:         Compute $N_i = N(\theta_t^{i,m}, A)$
14:         Send $N_i$ from each worker to coordinator
15:     **end for**
16:     Set $\theta_{t+1}^m = \theta_t^m + \alpha \frac{1}{n\sigma} \sum_{i=1}^n N_i \epsilon_i$
17:     Compute $b(\pi_{\theta_{t+1}^m})$
18:     Add $b(\pi_{\theta_{t+1}^m})$ to $A$
19: **end for**

---
**Algorithm 2** NSR-ES
---
1: **Input:** learning rate $\alpha$, noise standard deviation $\sigma$, number of policies to maintain $M$, iterations $T$, behavior characterization $b(\pi_\theta)$
2: **Initialize:** $M$ sets of randomly initialized policy parameters $\{\theta_0^1, \theta_0^2, ..., \theta_0^M\}$, archive $A$, number of workers $n$
3: **for** $j = 1$ **to** $M$ **do**
4:     Compute $b(\pi_{\theta_0^j})$
5:     Add $b(\pi_{\theta_0^j})$ to $A$
6: **end for**
7: **for** $t = 0$ **to** $T - 1$ **do**
8:     Sample $\theta_t^m$ from $\{\theta_t^0, \theta_t^1, \ldots, \theta_t^M\}$ via eq. [1]
9:     **for** $i = 1$ **to** $n$ **do**
10:         Sample $\epsilon_i \sim \mathcal{N}(0, \sigma^2 I)$
11:         Compute $\theta_t^{i,m} = \theta_t^m + \epsilon_i$
12:         Compute $b(\pi_{\theta_t^{i,m}})$
13:         Compute $N_i = N(\theta_t^{i,m}, A)$
14:         Compute $F_i = f(\theta_t^{i,m})$
15:         Send $N_i$ and $F_i$ from each worker to coordinator
16:     **end for**
17:     Set $\theta_{t+1}^m = \theta_t^m + \alpha \frac{1}{n\sigma} \sum_{i=1}^n \frac{N_i + F_i}{2} \epsilon_i$
18:     Compute $b(\pi_{\theta_{t+1}^m})$
19:     Add $b(\pi_{\theta_{t+1}^m})$ to $A$
20: **end for**

followed by 1 fully-connected layer with 256 hidden units, followed by a linear output layer with one neuron per action. The action space dimensionality can range from 3 to 18 for different games. ReLU activations are placed between all layers, right after virtual batch normalization units [53]. Virtual batch normalization is equivalent to batch normalization [54], except that the layer normalization statistics are computed from a reference batch chosen at the start of training. In our experiments, we collected a reference batch of size 128 at the start of training, generated by random agent gameplay. Without virtual batch normalization, Gaussian perturbations to the network parameters tend to lead to single-action policies. The lack of action diversity in perturbed policies cripples learning and leads to poor results [24].

The preprocessing is identical to that in Mnih et al. [7]. Each frame is downsampled to 84x84 pixels, after which it is converted to grayscale. The actual observation to the network is a concatenation of 4 subsequent frames and actions are executed with a *frameskip* of 4. Each episode of training starts with up to 30 random, no-operation actions (no-ops). Policies are also evaluated using a random number (sampled uniformly from 1-30) of no-op starts, whereas Mnih et al. [7] evaluates policies using starts randomly sampled from the initial portion of human expert trajectories (a dataset we do not have access to).

For all experiments, we fixed the training hyperparameters for fair comparison. Each network is trained with the Adam optimizer [55] with a learning rate of $\eta = 10^{-2}$ and a noise standard deviation of $\sigma = 0.02$. The number of samples drawn from the population distribution each generation was

**Algorithm 3** NSRA-ES

1: **Input:** learning rate $\alpha$, noise standard deviation $\sigma$, number of policies to maintain $M$, iterations $T$, behavior characterization $b(\pi_\theta)$
2: **Initialize:** $M$ sets of randomly initialized policy parameters $\{\theta_0^1, \theta_0^2, ..., \theta_0^M\}$, archive $A$, number of workers $n$, initial weight $w$, weight tune frequency $t_w$, weight delta $\delta_w$
3: **for** $j = 1$ **to** $M$ **do**
4:      Compute $b(\pi_{\theta_0^j})$
5:      Add $b(\pi_{\theta_0^j})$ to $A$
6: **end for**
7: $f_{best} = -\infty$
8: $t_{best} = 0$
9: **for** $t = 0$ **to** $T - 1$ **do**
10:      Sample $\theta_t^m$ from $\{\theta_t^0, \theta_t^1, \ldots, \theta_t^M\}$ via eq. 1
11:      **for** $i = 1$ **to** $n$ **do**
12:          Sample $\epsilon_i \sim \mathcal{N}(0, \sigma^2 I)$
13:          Compute $\theta_t^{i,m} = \theta_t^m + \epsilon_i$
14:          Compute $b(\pi_{\theta_t^{i,m}})$
15:          Compute $N_i = N(\theta_t^{i,m}, A)$
16:          Compute $F_i = f(\theta_t^{i,m})$
17:          Send $N_i$ and $F_i$ from each worker to coordinator
18:      **end for**
19:      Set $\theta_{t+1}^m = \theta_t^m + \alpha \frac{1}{n\sigma} \sum_{i=1}^{n} w * N_i \epsilon_i + (1 - w) * F_i \epsilon_i$
20:      Compute $b(\pi_{\theta_{t+1}^m})$
21:      Compute $f(\theta_{t+1}^m)$
22:      **if** $f(\theta_{t+1}^m) > f_{best}$ **then**
23:          $w = min(1, w + \delta_w)$
24:          $t_{best} = 0$
25:          $f_{best} = f(\theta_{t+1}^m)$
26:      **else**
27:          $t_{best} = t_{best} + 1$
28:      **end if**
29:      **if** $t_{best} \geq t_w$ **then**
30:          $w = max(0, w - \delta_w)$
31:          $t_{best} = 0$
32:      **end if**
33:      Add $b(\pi_{\theta_{t+1}^m})$ to $A$
34: **end for**

$n = 5000$. For NS-ES, NSR-ES, and NSRA-ES we set $M = 3$ as the meta-population size and $k = 10$ for the nearest-neighbor computation, values that were both chosen through an informal hyperparameter search. We lowered $M$ because the Atari network is much larger and thus each generation is more computationally expensive. A lower $M$ enables more generations to occur in training. We trained ES, NS-ES, NSR-ES, and NSRA-ES for a the same number of generations $T$ for each game. The value of $T$ varies between 150 and 300 depending on the number of timesteps per episode of gameplay (i.e. games with longer episodes are trained for 150 generations and vice versa). The figures in SI Sec. 6.9 show how many generations of training occurred for each game.

### 6.7 Humanoid Locomotion problem training details

The domain is the MuJoCo Humanoid-v1 environment in OpenAI Gym [36]. In it, a humanoid robot receives a scalar reward composed of four components per timestep. The robot gets positive reward for standing and velocity in the positive $x$ direction, and negative reward for ground impact energy and energy expended. These four components are summed across every timestep in an episode to get the total reward. Following the neural network architecture outlined by Salimans et al. [24], the neural network is a multilayer perceptron with two hidden layers containing 256 neurons each, resulting in a network with 166.7K parameters. While small (especially in the number of layers) compared to many deep RL architectures, this network is still orders of magnitude larger than what NS has been tried with before. The input to the network is the observation space from the environment, which is a vector $\in \mathbb{R}^{376}$ representing the state of the humanoid (e.g. joint angles, velocities) and the output of the network is a vector of motor commands $\in \mathbb{R}^{17}$ [36].

For all experiments, we fixed the training hyperparameters for fair comparison. Each network was trained with the Adam optimizer [55] with a learning rate of $\eta = 10^{-2}$ and a noise standard deviation of $\sigma = 0.02$. The number of samples drawn from the population distribution each generation was $n = 10000$. For NS-ES, NSR-ES, and NSRA-ES we set $M = 5$ as the meta-population size and $k = 10$ for the nearest-neighbor computation, values that were both chosen through an informal hyperparameter search. We trained ES, NS-ES, NSR-ES, and NSRA-ES for the same number of

generations $T$ for each game. The value of $T$ is 600 for the Humanoid Locomotion problem and 800 for the Humanoid Locomotion with Deceptive Trap problem.

## 6.8 Humanoid Locomotion problem tabular results

| ENVIRONMENT | ES | NS-ES | NSR-ES | NSRA-ES |
|---|---|---|---|---|
| ISOTROPIC | **8098.5** | 6010.5 | 6756.9 | 7923.1 |
| DECEPTIVE | 5.3 | 16.5 | 31.2 | **37.1** |

Table 2: **Final results for the Humanoid Locomotion problem.** The reported scores are computed by taking the median over 10 independent runs of the rewards of the highest scoring policy per run (each of which is the mean over ~30 evaluations).

## 6.9 Plots of Atari learning across training (generations)

Figure 5: **Comparison of ES, NS-ES, NSR-ES, and NSRA-ES learning on 12 Atari games.**

## 6.10 Overhead plots of agent behavior on the Humanoid Locomotion with Deceptive Trap Problem.

Figure 6: **Overhead plot of ES (left) and NS-ES (right) across 10 independent runs on the Humanoid Locomotion with Deceptive Trap problem.**

Figure 7: **Overhead plot of NSR-ES (left) and NSRA-ES (right) across 10 independent runs on the Humanoid Locomotion with Deceptive Trap problem.**