[Reviews · NeurIPS 2018]

Reviewer 1



Two heuristic mechanisms from neuroevolution study have been imported into the recently proposed evolution strategy for deep reinforcement learning. One is Novelty Search (NS), which aims to bias the search to have more exploration. It try to explore previously unvisited areas in the space of behavior, not in the space of policy parameters. The other is to maintain multiple populations in a single run. The authors proposed three variation of the evolution strategy combining these mechanisms. From the simulation results reported in the paper, the proposed method, NSRA-ES, doesn't seem to have a strong disadvantages over the original ES, while it improves its performance in some situations, though the improvements appear only after relatively large number of iterations. If so, one might want to simply restart the algorithm with different initialization. It is not very fair to compare the final performance after a sufficient number of iterations for the proposed algorithm. The authors say (at the first sentence of sec. 2.2) that optimizing for reward only can often lead an agent to local optima, and NS avoids deception. Please provide an evidence / reference. In many situation the state transition is stochastic and hence the reward is noisy. Then, the estimated gradient is hugely affected by this noise, and the signal-to-ratio will drop at some point. This can be one of the reason that a gradient based algorithm stacks. If so, I am curious to know if the proposed mechanisms are really more effective than just setting a greater sigma (standard deviation of the Gaussian noise to produce a parameter vector) in the ES. See for example, Arnold, D. V. (2006). Weighted multirecombination evolution strategies, TCS, vol. 361, 18–37. The sentence "NS-ES is still able to consistentlly solve the problem despite ignoring the environment's multi-part reward function". I feel this result is quite artificial, and without the modification of the environment done in this paper the result will be different. The authors modification should be very helpful for NS. Otherwise the diverse BC will simply result in searching in a wrong direction. (Rebuttal) A discussion about the noise standard deviation will be rather helpful to understand the goodness of the proposed method compared with the original algorithm with a tuned parameter. I appreciate that the authors will add an argument about it in the revised paper.

Reviewer 2



The paper presents the idea of combining Novelty Search with Evolutionary Strategies to overcome ES getting stuck in local optima for some subset of domains / problems. The idea is original as far as I am aware of. The authors give a good background about ES and NS topics. They introduce 3 different versions of combining NS and ES. They test the algorithms in simulated humanoid locomotion and some atari games. The proposed method works for domains where ES gets stuck in a local optima. To create such an environment the authors create a U shaped wall in locomotion problem. At the same time, they use behavior characterization that encourages moving to different spots. For this selected task and the selected behavior characterization using NS clearly makes sense. On the other hand, my high level concerns are: - The problem has to contain some "obstacle" that would get ES stuck at a local optima. - As the authors state, it might be possible to overcome the local optima by tuning hyper-parameters (line 309). Then ES might not benefit from NS anymore. Can the authors quantify the subset of domains where the proposed algorithm can solve the problem that cannot be solved by hyperparameter tuning. - behavior characterization function has to be selected carefully so that diversity can get ES unstuck. For locomotion problem, this makes sense. But it might not be possible to select the right function for more complex tasks. The paper is clearly written and easy to read. The presentation of the results in Figure 1 are hard to read. The fonts are too small, and the data plots are hard to distinguish in black and white. Figure 2 is missing the labels for x and y axis. I think that it would be better to include error bars in Figure 3. Otherwise the significance of the results are questionable (for some games the difference is less than 3%). Figure 3 should be split into a table (left) and a figure. The figure is hard to read and interpret in black and white. The idea is original as far as I am aware of. I like the idea of introducing diversity into ES. The first domain (human locomotion with U shaped wall) feels overcrafted for the proposed algorithm, but the atari application gives a better sense of how this algorithm could be adapted to different domains. As explained above the presentation of the results reduces the significance of the paper.

Reviewer 3



This submission proposes methods to enhance exploration in the Evolution Strategies framework introduced by Salimans et al. (ES), a black box optimization method that can leverage massive hardware resources to speedup training of RL agents, by introducing novelty seeking objectives from evolution/genetic algorithms literature. The balance between exploration and exploitation is a fundamental problem in RL, which has been thoroughly studied for many DRL methods, and is very interesting to explore it for alternative methods like ES. I find NSRA-ES particularly interesting, as it naturally handles the trade-off between exploration and exploitation, and will likely motivate new research in this direction. One of the weaknesses of the approach is the definition of the behavior characterization, which is domain-dependent, and may be difficult to set in some environments; however, the authors make this point clear in the paper and I understand that finding methods to define good behavior characterization functions is out of the scope of the submission. The article is properly motivated, review of related work is thorough and extensive experiments are conducted. The methods are novel and their limitations are appropriately stated and justified. The manuscript is carefully written and easy to follow. Source code is provided in order to replicate the results, which is very valuable to the community. Overall, I believe this is a high quality submission and should be accepted to NIPS. Please read more detailed comments below: - The idea of training M policies in parallel is somewhat related to [Liu et al., Stein Variational Policy Gradient], a method that optimizes for a distribution of M diverse and high performing policies. Please add this reference to the related work section. - The update of w in NSRA-ES somehow resembles the adaptation of parameter noise in [Plapper et al., “Parameter Space Noise for Exploration”, ICLR 2018]. The main difference is that the adaptation in Plappert et al. is multiplicative, thus yielding more aggressive changes. Although this is not directly compatible with the proposed method, where w is initialized to 0, I wonder whether the authors tried different policies for the adaptation of w. Given its similarity to ES (where parameter noise is used for structured exploration instead of policy optimization), I believe this is a relevant reference that should be included in the paper as well. - It seems that the authors report the best policy in plots and tables (i.e. if f(theta_t) > f(theta_{t+k}), the final policy weights are theta_t). This is different to the setup by Salimans et al. (c.f. Figure 3 in their paper). I understand that this is required for methods that rely on novelty only (i.e. NS-ES), but not for all of them. Please make this difference clear in the text. - Related to the previous point, I believe that section 3.1 lacks a sentence describing how the final policy is selected (I understand that the best performing one, in terms of episodic reward, is kept). - In the equation between lines 282 and 283, authors should state how they handle comparisons between episodes with different lengths. I checked the provided code and it seems that the authors pad the shorter sequence by replicating its last state in order to compare both trajectories. Also, the lack of a normalization factor of 1/T makes this distance increase with T and favors longer trajectories (which can be a good proxy for many Atari games, but not necessarily for other domains). These decisions should be understood by readers without needing to check the code. - There is no caption for Figure 3 (right). Despite it is mentioned in the text, all figures should have a caption. - The blue and purple color in the plots are very similar. Given the small size of some of these plots, it is hard to distinguish them -- especially in the legend, where the line is quite thin. Please use different colors (e.g. use some orangish color for one of those lines). - Something similar happens with the ES plot in Figure 2. The trajectory is quite hard to distinguish in a computer screen. - In SI 6.5, the authors should mention that despite the preprocessing is identical to that in Mnih et al. [7], the evaluation is slightly different as no human starts are used. - In SI 6.6, the network description in the second paragraph is highly overlapping with that in the first paragraph. ---------------------------------------------------------------------------------------------------------------------------------------------------------- Most of my comments had to do with minor modifications that the authors will adress. As stated in my initial review, I vote for accepting this submission.